# TLR/WNT: A Novel Relationship in Immunomodulation of Lung Cancer

**DOI:** 10.3390/ijms23126539

**Published:** 2022-06-11

**Authors:** Aina Martín-Medina, Noemi Cerón-Pisa, Esther Martinez-Font, Hanaa Shafiek, Antònia Obrador-Hevia, Jaume Sauleda, Amanda Iglesias

**Affiliations:** 1Instituto de Investigación Sanitaria de les Illes Balears (IdISBa), 07120 Palma, Spain; ceronpisa.n@gmail.com (N.C.-P.); esther.martinez@ssib.es (E.M.-F.); antonia.obrador@ssib.es (A.O.-H.); jaume.sauleda@ssib.es (J.S.); 2Medical Oncology Department, Hospital Universitario Son Espases, 07120 Palma, Spain; 3Chest Diseases Department, Faculty of Medicine, Alexandria University, Alexandria 21526, Egypt; whitecoat.med@gmail.com; 4Molecular Diagnosis Unit, Hospital Universitario Son Espases, 07120 Palma, Spain; 5Department of Respiratory Medicine, Hospital Universitario Son Espases, 07120 Palma, Spain; 6Centro de Investigación Biomédica en Red in Respiratory Diseases (CIBERES), 28029 Madrid, Spain

**Keywords:** TLR, WNT, lung cancer

## Abstract

The most frequent cause of death by cancer worldwide is lung cancer, and the 5-year survival rate is still very poor for patients with advanced stage. Understanding the crosstalk between the signaling pathways that are involved in disease, especially in metastasis, is crucial to developing new targeted therapies. Toll-like receptors (TLRs) are master regulators of the immune responses, and their dysregulation in lung cancer is linked to immune escape and promotes tumor malignancy by facilitating angiogenesis and proliferation. On the other hand, over-activation of the WNT signaling pathway has been reported in lung cancer and is also associated with tumor metastasis via induction of Epithelial-to-mesenchymal-transition (EMT)-like processes. An interaction between both TLRs and the WNT pathway was discovered recently as it was found that the TLR pathway can be activated by WNT ligands in the tumor microenvironment; however, the implications of such interactions in the context of lung cancer have not been discussed yet. Here, we offer an overview of the interaction of TLR-WNT in the lung and its potential implications and role in the oncogenic process.

## 1. Introduction

Lung cancer is the most frequent cause of death by cancer worldwide [1]. The 5-year survival rate in lung cancer varies from 73% (in stage IA) to 13% (in stage IV). Lung cancer can be classified into two main subtypes according to the histological pattern, namely, small-cell lung carcinomas (SCLC) and non-small cell lung carcinomas (NSCLC) [2]. NSCLC represents 85% of lung cancer cases where adenocarcinoma (ADC), squamous cell carcinoma (SCC), and large-cell carcinoma (LCC) are the main NSCLC subtypes [3]; while SCLC represents 15% of all lung cancers and is classified as small cell carcinoma when it occurs without a combination of large cell carcinoma, that is, with pure histology [4]. Both for the development of mammals, as well as for the regeneration and repair of tissues, the regulation of cell proliferation and survival is very important. What leads to cancer is cell proliferation and survival [5]. In 1863, Rudolf Virchow was the first to identify that leukocytes were present in tumor tissue, and thus he hypothesized that there was a link between inflammation and tumor progression [6]. Host immune responses must induce cell proliferation and survival, which is necessary for host defense after infection, as they stimulate leukocyte proliferation in the bone marrow and induce clonal expansion of lymphocytes. Innate immunity is the first line of defense against microorganisms, it is characterized by not being clonal, and it acts on a very large number of pathogens since it focuses on the recognition of a few conserved structures [7]. Immune cells are capable of mediating pathogen recognition through the expression of the so-called pattern recognition receptors (PRRs), such as the Toll-like receptors (TLRs), which are responsible for the identification of pathogen-associated molecular patterns (PAMPs) and the induction of the inflammatory response [8]. Furthermore, immune-mediated inflammatory responses through TLRs can indirectly initiate tissue repair processes after local infection [5]. The WNT (Wingless/Integrated) pathway is a key developmental pathway involved in tissue repair and regeneration. Upon specific binding to its ligands, the WNT pathway is activated and drives the translocation of transcription factors that regulate gene expression [9]. The role that the TLR family and the WNT signaling pathway play in cancer development and progression has been largely explored in the literature, and alterations of both pathways have been independently reported in lung cancer. However, although an interaction between both pathways exists, there is little reported information about the link between TLR and WNT in lung cancer. In this review, we will summarize the current knowledge of TLR and WNT pathways in lung cancer with a focus on the interaction of both signaling pathways and their involvement in tumor development and progression.

## 2. Toll-like Receptors (TLRs) in Lung

### 2.1. TLRs and TLR Signaling

TLRs comprise a family of receptors in the immune system, which are structurally characterized by extracellular leucine-rich repeats (LRRs) and an intracellular Toll/IL-1 receptor (TIR) signaling domain [10]. The TLR family consists of ten members in humans (TLR1-TLR10) [8]. The presence of mRNA or protein of TLR1, TLR2, TLR3, TLR4, TLR7, TLR8, TLR9, and TLR10 has been detected in lung tissue [11,12]. TLRs are activated following the direct recognition of a wide range of pathogen-associated molecular patterns (PAMPs) that include bacterial lipopolysaccharides (LPS), lipoproteins, and flagellin, as well as viral and bacterial nucleic acids [8]. Once PAMPs are recognized by TLRs, a signaling cascade is initiated that triggers the expression of inflammatory mediators, including many cytokines, chemokines, and cell adhesion molecules [13]. In addition to PAMPs, TLRs can also detect the so-called DAMPs (damage-associated molecular patterns), which are a variety of endogenous molecules that are released as a result of cell or tissue damage. Similar to PAMPs, DAMPs can also trigger inflammation even in the absence of infection [14] (Figure 1). In addition to their role in mediating the inflammatory response, TLRs can also regulate both proliferation [15] and cell survival by suppressing apoptosis [5,16,17], thus protecting against injury and initiating tissue repair. However, if this function becomes dysregulated, it can lead to uncontrolled proliferation and tumor development [18]. In fact, different tumor cells seem to express their own set of TLRs, suggesting a role for TLRs in the regulation of tumor growth [19]. Therefore, TLRs act as a double-edged sword where, on the one hand, they identify cancer-specific antigens and activate innate responses, which can serve as a tumor suppressor, and on the other hand, they can promote cell proliferation and survival and latent chronic inflammation by inducing persistent, adaptive responses that facilitate the process of tumorigenesis [20,21,22].

### 2.2. TLRs Role in Lung Cancer

At present, approximately 18% of cancer cases are caused by infection and chronic inflammation, and patients with chronic inflammatory diseases have a higher risk of cancer development [18,23]. In the tumor cells or within the tumor microenvironment, TLRs can promote carcinogenesis through various mechanisms such as proinflammatory, prosurvival, proliferative, and immunosuppression [18]. In the tumor cells, aberrant activation of TLR might occur, thus increasing NF-κB activity, which upregulates pro-apoptotic genes and inhibits JNK-mediated pro-apoptotic signals, therefore favoring tumor survival [24]. The paracrine tumoral effects are basically mediated by the cells of the immune system via TLR-released cytokines that will suppress the T cell response, thus promoting immunoevasion by the tumor (Figure 2, Panel A). TLRs can also activate immune cells, producing a cytokine storm in response to the tumor microenvironment and to the DAMPs derived from cell or tissue damage as a result of harmful processes such as chemotherapy or radiotherapy [25]. In addition, in the context of the tumor, the cytokines released by immune cells can have a paracrine effect on the tumor cells, which causes them to secrete proteins that help keep the cytokine storm ongoing [26]. Moreover, Choi et al. highlighted that tumors associated with overexpression of TLRs result in a poor prognosis for cancer patients [27]. In fact, tumor-derived DAMP proteins may have an effect on the endothelial-to-mesenchymal transition (EMT) mechanism, which is responsible for tumor metastasis [28,29], and can also bind to TLRs on tumor cells activating signaling pathways involved in tumor migration, invasion, and metastasis [30,31]. Activation of TLRs in the tumor cell induces the production of proinflammatory cytokines IL-6 and IL-8, a strong neutrophil chemoattractant, and vascular endothelial-derived growth factor (VEGF), which is a key factor in promoting angiogenesis [32,33] (Figure 2, Panel B). 

The role of IL-6 in lung cancer needs to be taken carefully. Despite the fact that the major role of IL-6 in cancer seems to be protumoral by promoting proliferation, survival, angiogenesis, metastasis, and immunoevasion [34], a new role in inducing anti-tumor immunity has begun to be explored as it appears to help promote the anti-tumoral effects of the T cells although the mechanisms are not yet fully understood [35,36].

However, besides its role in promoting tumor growth, it is known that TLRs are also capable of inducing tumor-suppressive effects, or even a dual role has been proposed for some of the TLRs depending on the context. For instance, TLR5 has been reported to exert an anti-tumoral effect due to its role in the immune response through the regulation of dendritic cells (DC), which can kill tumor cells [37]. On the other hand, some TLRs seem to play a dual role in promoting or suppressing tumor progression depending on the type of cancer; for example, TLR7/8 has been reported to mediate anti-tumor effects due to their role on DC and NK cell activation [38,39]. The role of TLRs in mediating pro- or anti-tumoral effects in the context of each cancer type is summarized in Table 1.

In the lung, the studies so far rather support a role for TLR signaling in promoting the growth and survival of malignant cells. Sureshbabu et al. observed that the release of TGF-β1 by lung epithelial cells facilitates both apoptosis and inflammation [40]. In addition, activation of TLR2 and TLR4 induce extracellular matrix remodeling and Epithelial Growth Factor Receptor (EGFR)-mediated signaling, respectively, stimulating lung carcinoma progression [41]. Furthermore, He et al. showed that stimulation of human lung cancer cells with functional TLR4 resulted in a secretion of immunosuppressive cytokines such as VEGF, TGF-β, and IL-8, which induced resistance to TNFα-induced apoptosis and TRAIL in vitro [42]. TLR7 has been reported to induce cell survival and tumor growth [43], most likely via NF-κB activation and upregulation of *Bcl-2* [44]. Droemann et al. observed a high TLR9 expression in both primary lung cancer specimens and tumor cell lines. The activation of the TLR9 pathway resulted in the production of monocyte chemoattractant protein-1 () and the reduction in TNFα-induced apoptosis [45]. 

## 3. WNT in Lung Cancer

### 3.1. WNT Signaling Pathway

The WNT (Wingless/Integrated) pathway is a developmental pathway involved in repair and regeneration processes, among others, activated by the WNT ligands (19 in humans), which are evolutionarily conserved glycoproteins that are secreted and bind to cell surface receptors termed Frizzled (Fzd). To date, 10 Fzd receptors have been identified in mice and humans [59]. Depending on the context, WNT ligands can initiate at least three different intracellular signaling cascades: the canonical WNT/β-catenin pathway, and the non-canonical WNT/β-catenin pathway (β-catenin-independent pathway), which further divides into the planar cell polarity (PCP), and the WNT/Ca2+ pathway [60,61]. In mammalian cells, the canonical WNT/β-catenin pathway is activated when the WNT ligand binds to the Fzd receptor and the low-density lipoprotein receptor-related protein 5/6 (LRP5/6). Activation of the pathway and recruitment of Axin and associated proteins (GSK3β, CK1α, APC) to Fz-LRP5/6 in the membrane negatively regulates the β-catenin destruction complex. β-catenin can then translocate to the nucleus to act as a transcription factor for genes that encode proteins involved in cell proliferation, differentiation, and stem cell maintenance [62] (Figure 3, panel A). In the non-canonical PCP pathway, WNT ligands bind to Fzd receptors and the RAR-related orphan receptor (ROR) and signals to the cytoplasmic phosphoprotein Dishevelled (DVL), which activates Rho-associated (ROCK) or the JUN-N-terminal kinase (JNK), thus inducing gene expression by the activator protein-1 (AP-1) [63] involved in cytoskeletal rearrangement, cell motility, and coordinated cell polarity (Figure 3, panel B). 

In the non-canonical WNT calcium-dependent pathway, WNT ligands bind Fzd and activate phospholipase C (PLC), which hydrolyzes certain lipids, resulting in the release of intracellular calcium and the activation of the nuclear factor of activated T cells (NFAT), which moves to the nucleus and regulates the expression of target genes involved in embryonic patterns and tissue homeostasis [63]. (Figure 3, panel C). The WNT pathway is essential for the proper development of organs and tissues, including the lungs, and also regulates key repair processes in both physiological and pathological situations [64]. Aberrant regulation of the WNT signaling pathway has been a recurrent issue in cancer research [65] due to its contribution to tumorigenesis and metastasis.

### 3.2. Alterations of the WNT Signaling Pathway in Lung Cancer

The WNT pathway is involved in the maintenance of progenitor stem cells in epithelial tissues, including the lung, playing a major role in mediating processes involved in homeostasis, regeneration, and repair [68,69]. Carcinomas arising from these tissues often exhibit aberrant activation of the WNT pathway by mechanisms such as mutations in APC, β-catenin, or axin [70] and, more recently, through autocrine activation of WNT [71]. At the level of WNT ligands, overexpression of WNT1, WNT3, WNT5A, and WNT11, along with the Fzd8 receptor, has been reported in NSCLC [72,73,74,75]. Moreover, high levels of WNT1, WNT5A [76], and WNT3 [73] in NSCLC patients were correlated with lower survival. Further, WNT3 was observed to promote cell invasion and independent growth of tumor cells, and WNT3A treatment of NSCLC cells facilitated the EMT process, increasing the like-hood of NSCLC metastasis [77]. In fact, overexpression of Glypican-5, which traps WNT3A competing with its Fzd receptor, prevents EMT and metastasis in lung adenocarcinoma [78]. WNT5A ligand expression is highly increased in NSCLC and has been associated with poor prognosis [79,80]. Another study reported a differential overexpression of WNT ligands depending on the NSCLC subtype since overexpression of non-canonical WNT5A was observed in squamous cell carcinomas while canonical WNT7B was found in adenocarcinomas [81]. On the other hand, WNT7A and its Fzd9 receptor were found downregulated in lung tissue from NSCLC patients, most likely preventing the activation of JNK downstream of the non-canonical WNT pathway, which maintains epithelial cell differentiation and suppresses tumor cell proliferation [82]. The O-acyltransferase Porcupine is required in WNT-producing cells for proper ligand maturation and secretion [83], and it was found overexpressed in lung adenocarcinoma tissue [84]. Of notice, WNT ligands were discovered recently to be transported through extracellular vesicles (such as exosomes) in a Porcupine-dependent manner, which are membranous vesicles that mediate intercellular communication by transporting specific cargos (i.e., proteins, lipids, and nucleic acids) between the producer and the receiver cells [85]. It is known that tumor-derived exosomes mediate important roles in tumor progression [86]; in fact, a pro-tumoral role of exosomal WNT5A has been reported in several cancers [87,88].

At the level of the WNT signaling cascade, activation of WNT/β-catenin seems to be a signature in Kras positive lung tumors that predicts the response to pharmacologic treatment and is correlated with faster disease progression [89,90]. The WNT/β-catenin pathway has also been associated with metastasis of NSCLC tumors via induction of key transcription factors (*TCF, LEF1*, and *HOXB9*) that will favor EMT-like processes [90,91].

Abnormalities in key pathway modulators were also found to induce aberrant WNT activation. For instance, loss of function mutations in Fzd inhibitors Rnf47 and Znrf3 are present in lung cancer [92], and specific mutation in *Lrp6* was found to increase the risk of NSCLC in smokers [93]. Tobacco smoke was also found to induce the secretion of the WNT agonist Dkk1 producing a protumoral effect [94]. *Ror1* is upregulated in lung cancer and has been associated with resistance to Tyrosine Kinase Inhibitor (TKIs) therapy in epidermal growth factor receptor (EGFR) positive NSCLC [95,96]. 

## 4. TLR/WNT Crosstalk

### 4.1. TLR/WNT Crosstalk in the Lung

Recent studies suggest that the WNT/β-catenin and NF-κB signaling pathways cross-regulate each of their activities and functions [97], and accumulating evidence has shown that a negative regulation of immune cells such as T regulatory cells (Treg) as well as the reprogramming of dendritic cells (DC) for immunotolerance is due to the fact that inflammatory responses are capable of activating the WNT/β-catenin pathway [98,99]. 

A special relationship between TLRs and WNT signaling happens through the non-canonical WNT5A ligand. *WNT5A* transcription is regulated by many proteins, including nuclear factor NF-κB. Katoh et al. found that within the B region of the *WNT5A* promoter, there was a conserved NF-κB binding site. This explains the mechanisms by which *WNT5A* are upregulated through TNFα and TLR signals [100]. In the last decade, Trinath et al. identified a novel role for the WNT signaling pathway by observing cross-regulation with TLRs in macrophages where the TLR proinflammatory signal is downregulated by the WNT5A ligand. In more detail, they found that upon microbial challenge, activation of dectin-1 (a receptor present in macrophages) stabilizes β-catenin and upregulates non-canonical WNT5A secretion, which mediates downregulation of MyD88 (a major effector of the TLR pathway), thus resulting in decreased expression of IL-12, IL1-β and TNF-α [101]. In fact, Mehmeti et al. discovered a novel role of WNT5A in inflammatory cells by direct binding to TLR2/4, inducing L-10 expression and immune tolerance [102]. Upon infection by mycobacteria, WNT5A was induced in antigen-presenting T cells in a TLR-NF-κB-dependent manner, where it mediates the secretion of IL-12 and IFN [103], and both WNT5A and WNT3A were found to inhibit TLR-induced secretion of proinflammatory cytokines in DC [98,104]. Another study by Neumann et al. also suggested a negative regulation of TLR by WNT signaling as they observed that proinflammatory mediators induced by TLR were increased while WNT/β-catenin signaling was decreased in macrophages following bacterial infection [98,104]. The negative regulation between both signals is also observed in lung disease. For instance, downregulation of the WNT/β-catenin pathway is found in the lung epithelium o COPD smokers, which are known to be susceptible to bacterial infection [80]. Moreover, in the context of infection by pathogens, it has been reported that the WNT/β-catenin pathway is constitutively activated in lung epithelial cells, thus mediating the recruitment of macrophages, specifically at the sites of bacterial infection [104,105]. In fact, TLR-4-MyD88 signaling is also associated with WNT5A to induce IL-12p40 and IL-6 expression in macrophages [106]. Similarly, LPS/IFN gamma induces WNT5A expression in macrophages, which involves the activation of TLR signaling and NF-κB by mediating the release of IL-1b, IL-6, IL-8, and MIP1b [107]. In contrast, in the context of viral infection, Oderup et al. found that WNT5A rather inhibits the production of IL-6 [98,104]. In addition, it was found that infected macrophages seem to undergo differentiation and proliferation in response to WNT6 [104,105]. Altogether, these studies support the role of WNT signaling in regulating the TLR inflammatory response in immune cells and in mediating immunotolerance and macrophages homing upon lung infection. 

### 4.2. TLR/WNT in Lung Cancer

As previously mentioned, TLRs and the WNT signaling have been observed to play a role in promoting carcinogenesis and tumor cell proliferation. Increased/aberrant WNT signaling in the tumor cell usually leads to a great release of WNT ligands and proinflammatory mediators (i.e., IL-6 and MCP-1), which is sustained over time due to a self-feeding loop. These mediators will then attract the immune cells to the tumor, which are susceptible to TLR activation through direct binding with the tumor-derived WNT ligands. TLR active monocytes would then secrete IL-10, thus inducing immunosuppression, which helps tumor cells evade the immune system (Figure 4). 

The WNT/β-catenin pathway components modulate inflammatory and immune responses via the interaction with NF-κB and, thus, significantly influence the progression of inflammation and cancer. In this line, Li et al. found that NF-κB-dependent and smoke-induced inflammation activates numerous growth pathways in lung cancer cells, including the WNT/β-catenin signaling pathway [108]. Moreover, in a recent analysis of the correlation between the expression of WNT ligands and 23 immunosuppressive genes across all cancer types in the TCGA dataset, high levels of WNT1 were found to significantly negatively correlate with CD8 + T cells, suggesting that it induces immune resistance in lung adenocarcinoma cells [109]. 

The WNT non-canonical WNT5A ligand has shown important roles in both innate and adaptive immunity to infections, being associated with several inflammatory processes such as induction of proinflammatory cytokines, regulation of migration, and recruitment of various immune effector cells, and inducement of T cell differentiation [110]. In cancer, WNT5A has mainly been implicated as an oncogenic protein that is involved in the invasion and metastasis of many cancers [103,111]. Tobacco smoke is a very potent inducer of lung cancer [112], and exposure to cigarette smoke-extract induces WNT5A expression in human bronchial epithelial cells [113] and increases the expression of TLR in macrophages [114]. In the lung, WNT5A is overexpressed, correlating with tumor aggressiveness, lower survival, and the development of resistance to therapy [80,115].

It is known that patients suffering from other chronic lung diseases where WNT and/or TLR play an important role are often at higher risk of developing lung cancer. For instance, an overactivation of the non-canonical WNT signal, mostly represented by a WNT5A overexpression, has been reported in alveolar epithelial cells from COPD patients, and inhibition of WNT5A results in attenuation and amelioration of elastase-induced emphysema in mice [116]. Upregulation of TLR signaling also takes place in COPD, where it seems to contribute to prolonging the oxidant/antioxidant imbalance [117], and overactivation of TLRs often happens in COPD due to common exacerbations caused by bacterial infections [118]. Idiopathic Pulmonary Fibrosis (IPF) has been largely related to the development of lung cancer [119]. Aberrant WNT activation and overexpression of WNT ligands (WNT3A in epithelium and WNT5A in fibroblasts) is a hallmark in IFP [120,121], and TLR also plays a role in disease since there is an ongoing inflammation due to an upregulation of proinflammatory cytokines IL-1 and IL-6 in part in response to an increased WNT signaling [122]. The fact that both WNT and TLR signals coexist not only in lung cancer but also in other lung diseases related to cancer highlights the prominent role of the interaction of these two signals in tumor development. 

## 5. Conclusions

Lung cancer is a frequent tumor worldwide and is associated with poor survival. TLRs and WNT signaling play important roles in the immune response to various inflammatory and tumorigenic processes (Figure 5). Various subtypes of TLRs and WNT signaling members have been involved in tumor cell proliferation, tumor invasion, metastasis, and poor survival, especially in NSCLC. In the context of lung cancer, autocrine WNT/TLR promotes tumor growth and survival and further recruits monocytes and mediates immunosuppressive effects that support tumor progression. More research is still required to assess the role of TLRs and WNT pathways in lung cancer and explore the possibility of dual therapeutic options. 

## Figures and Tables

**Figure 1 ijms-23-06539-f001:**
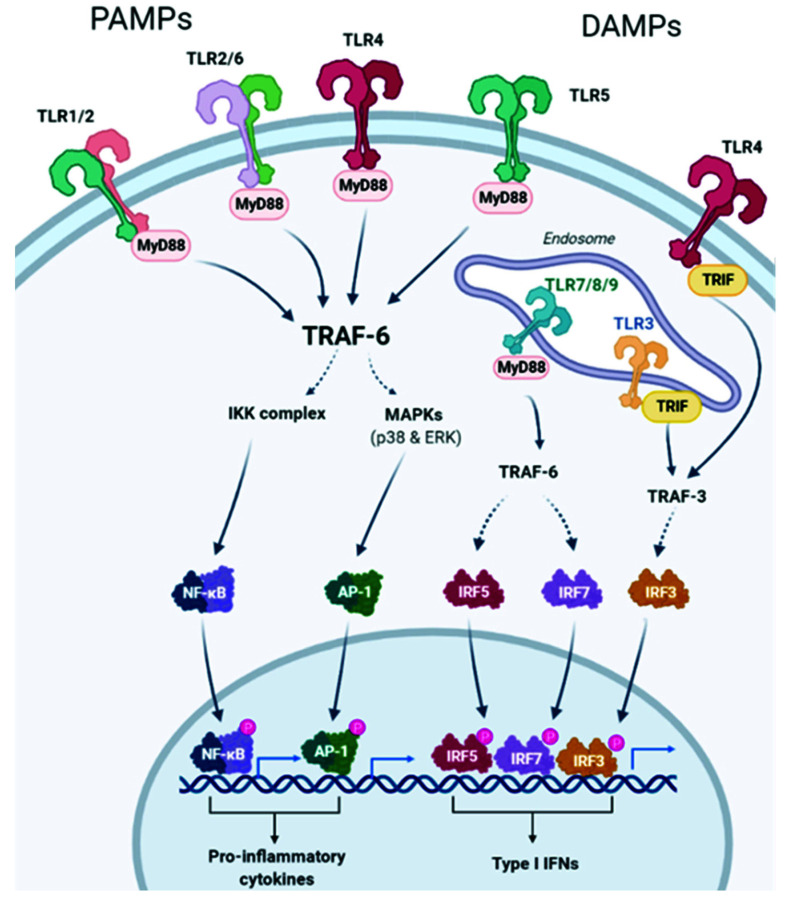
TLR2, TLR1, TLR4, TLR5, and TLR6 are expressed on the outer cell membrane; they recognize both extracellular pathogen-associated molecular patterns (PAMPs) and damage-associated molecular patterns (DAMPs). TLR3, TLR7, TLR8, and TLR9 are located in endosomal compartments where they bind to nucleic acids. All TLRs use the myeloid differentiation primary response protein 88 (MyD88) pathway, except TLR3, whose signaling depends on the IFN-α (TRIF) pathway that contains the TIR domain-containing adapter. TLR activates the MyD88-dependent canonical pathway that, through recruitment of TNF receptor-associated factors (TRAF) 6, leads to the activation of the transcription factor NF-KB, mitogen-activated protein kinase (MAPK), and activator protein-1 (AP-1), with the consequent induction of the production of proinflammatory cytokines. Intracellular TLRs are primarily involved in the type I interferon response. TLR7, TLR8, and TLR9 activate interferon regulatory factor 7 (IRF7) through the recruitment of TRAF6. TLR3 and TLR4 use the TRIF-dependent pathway. TRAF3 is activated and consequently activates IRF3, resulting in the induction of type I interferon (IFN) production. Created with BioRender.com (accessed on 24 February 2022).

**Figure 2 ijms-23-06539-f002:**
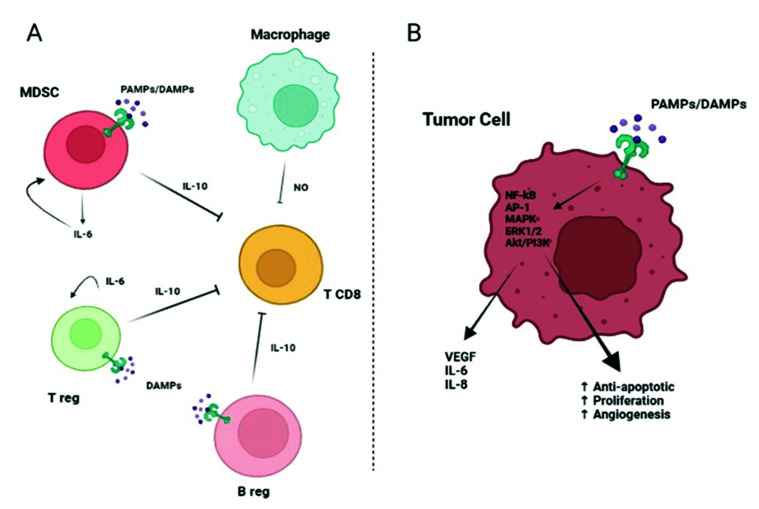
(**A**): TLR activation on myeloid-derived suppressor cells (MDSCs), B regulatory cells (Breg), and T regulatory cells (Treg) leads to an immunosuppressive effect. Indeed, MDSCs can induce macrophages to release nitric oxide (NO) that suppresses the activity of T CD8+ lymphocytes. Moreover, activated MDSCs, Bregs, and Tregs release IL-6 and IL-10 with a consequent synergic inhibitory effect on T CD8+ cells. (**B**): TLR expressed on tumor cells can bind DAMPs/PAMPs and activate intracellular signaling, such as nuclear factor kappa-light-chain-enhancer of activated B cells (NF-kB), AP-1, Akt/PI3K, extracellular signal-regulated kinase (ERK)1/2, and MAPK. The consequence is a pro-tumor effect due to the release of IL-6, IL-8, and VEGF and the promotion of proliferation, angiogenesis, and protection from apoptosis. Continuous arrows indicate activation or secretion. Created with BioRender.com (accessed on 4 May 2022).

**Figure 3 ijms-23-06539-f003:**
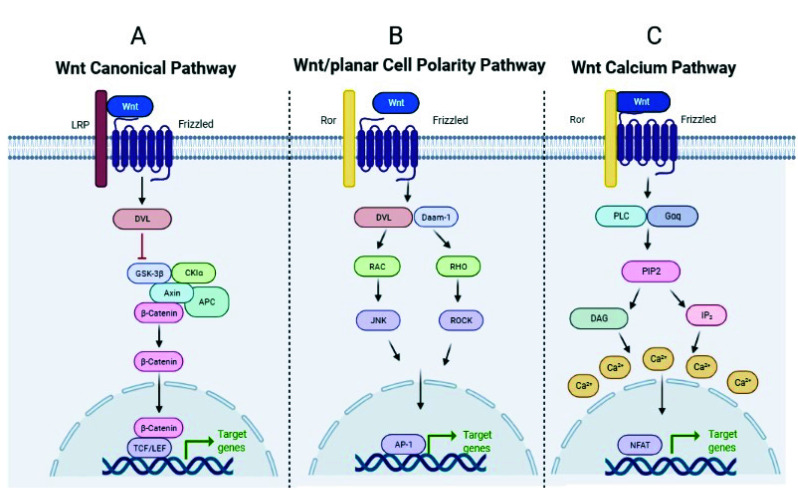
Scheme of the three WNT pathways. Created by BioRender.com and modified from Koni et al. [66] and Hiremath et al. [67].

**Figure 4 ijms-23-06539-f004:**
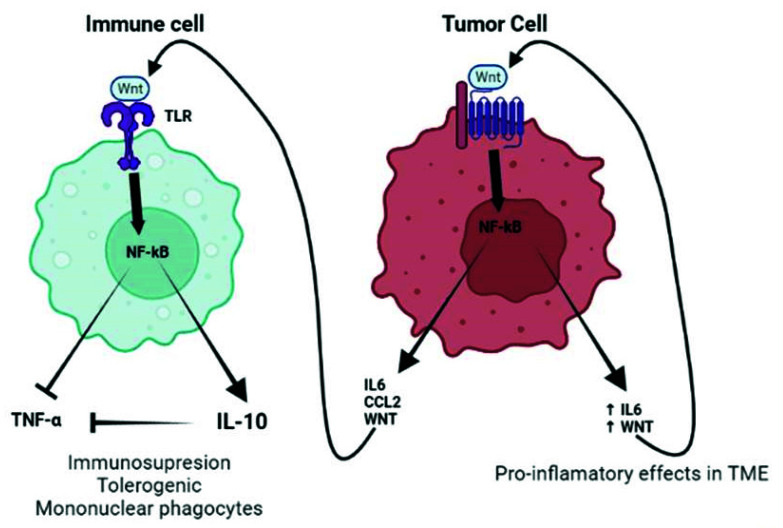
Interaction between tumor-derived WNT and immune cells. Increased production of WNT by tumor cells (right part of the figure) stimulates an autocrine pathway that promotes the secretion of proinflammatory cytokines (IL-6) and chemokines (MCP-1). Both cytokines and WNT are maintained at a high level of expression by a self-feeding loop. Chemokines are critical for populating the tumor microenvironment (TME) with immune cells. The different cytokines induced by WNT, together with the WNT ligand itself, would stimulate other cells in the TME, such as tumor-associated fibroblasts or endothelial cells that further amplify the effects of WNT. At a later stage, once immune cells are recruited to the TME, WNT begins to play a different role (left part of the figure). In this new scenario, WNT activates a TLR/MyD88/p50 pathway, promoting the synthesis of the anti-inflammatory cytokine IL-10. As a result, WNT induces immunosuppression and the formation of tolerogenic mononuclear phagocytes. Created with BioRender.com (accessed on 21 March 2022).

**Figure 5 ijms-23-06539-f005:**
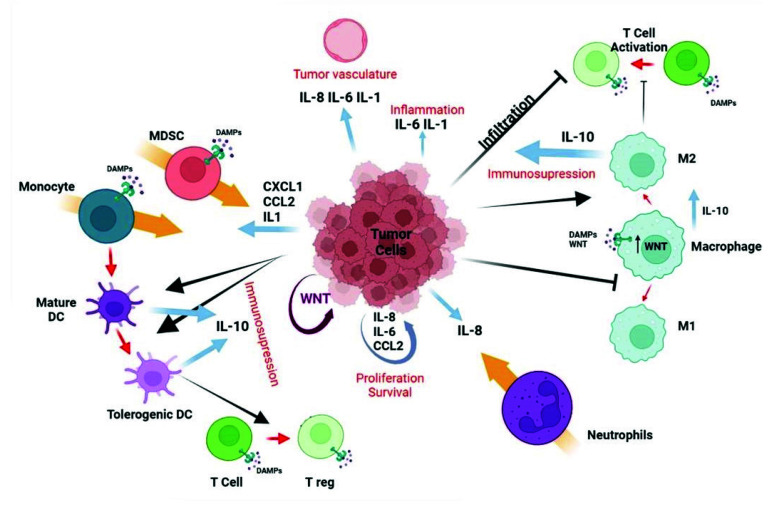
Immunomodulatory effects of WNT on the tumor microenvironment. The autocrine effect of WNT (garnet arrow) drives the secretion of various cytokines and chemokines by tumor cells (blue arrows), which in turn promote different processes (in red) and chemotaxis of various cell types (orange arrow). WNT also promotes IL-10 secretion by dendritic cells (DC) and macrophages. WNT released by tumor cells promotes (black lines with arrowheads) or inhibits (black lines) maturation and/or differentiation (red arrows) of immune cells. The effects associated with WNT are angiogenesis, inflammation, and immunosuppression, as well as proliferation and survival of tumor cells. Created with BioRender.com (accessed on 21 March 2022).

**Table 1 ijms-23-06539-t001:** Pro- or anti-tumoral effects mediated by TLRs.

Receptor	Tumor Type	Tumor Role	Mechanism	Pathways or Molecules Involved	Ref
TLRs	Intestine, liver, skin	Protumoral	Inflammation	NF-κB, IL-1β, TNFα and IL-6	[46]
TLR2	Gastric	Protumoral	Cell survival and proliferation in gastric tumor epithelium	PI3K/Akt, ERK1/2, JNK MAPKs, NF-kb	[47]
TLR2/6	Lewis lung carcinoma	Protumoral	Inflammation, macrophage activation, metastasis	Myd88, TRIF, ECM-protein versican, IL-6, TNF-α	[48]
TLR2/4	Lung	Protumoral	ECM remodeling,	EGFR	[38]
TLR3	Breast	Protumoral	Tumor growth	β- Catenin, NF-κB	[49]
TLR4	Liver	Protumoral	Inflammation	NF-κB	[50]
TLR4	Lung	Protumoral	immunosuppression, antiapoptosis	VEGF, TGF-β, IL-8	[42]
TLR4	Colon	Protumoral	Inflammation, tumor burden	Cox-2	[51]
TLR4	Skin	Protumoral	Inflammation	Myd88	[52]
TLR7	Pancreatic	Protumoral	Stromal inflammation	NF-kB, MAPKs	[53]
TLR7, TLR8	Lung	Protumoral	Inflammation, tumor cell growth and survival, chemoresistance	NF-kB, MAPK, IRF, *Bcl-2*, IL-6, IL-8, CSF-2, IL-1α, IL-12, NOS-2	[44]
TLR9	Lung	Protumoral	Monocyte recruitment, antiapoptosis	MCP1	[45]
TLR2	Lung	Antitumoral	Monocytic myeloid-derived suppressor cell	JNK, TNF-α, IL-12p40, IL-12p70	[54]
TLR3	Prostate, liver	Antitumoral	Treatment with agonist stops tumor growth	Type I interferons	[55]
TLR3	Liver	Antitumoral	Induction of tumor parenchyma (hepatocytes) cell death; induction of intratumor expression of chemokines that attract NK cells or T cells to the tumor microenvironment; and activation of tumor-infiltrating NK cells that promote cytotoxic activity.	IFN-γ	[56]
TLR5	Breast, colon	Antitumoral	Immune cell activation	Dendritic cells	[37,57,58]
TLR7/8	Melanoma	Antitumoral	Immune cell activation	Dendritic cells and Natural Killer	[38,39]

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
