# Peer review of "TLR/WNT: A Novel Relationship in Immunomodulation of Lung Cancer"

_ijms, 2022, doi:10.3390/ijms23126539_

Round 1

Reviewer 1 Report

General comments:

The authors of the present manuscript "TLR/WNT: a novel relationship in immunomodulation of lung cancer" set out to review the literature about this interesting topic. Unfortunately, they were overwhelmed by the vast literature, and were bogged down at certain points of their topic that bear little or no relevance to their final aim. Often, they repeat and shorten other reviews, which would have been better to refer to and focus the review.

Specific comments:

  1. The two compared categories TLR and WNT cannot be compared as it first mentioned in the text. WNT is a large signaling system, TLR is a receptor family in the immune system. The sentence needs rephrasing.
  2. Figure 3 should be A, B, and C panels, not left/medium/right.
    Also it would be better to refer to some good summary reviewes about WNT signaling. Keep the figure, but do not try to explain it in detail.
  3. It appears that Figure 3 was based on and redrawn from specific articles which should be referenced.
  4. The article would benefit from structured tables. What kind of molecules are induced by proinflammatory cytokines? "Other proteins" is just not enough, as it has become broadly known that extracellular vesicles play an important role in spreading information amongst cells, tissues or even organs. EV-s, however, contain far more than just proteins. miRNAs, lnRNAs, etc. What expression is triggered by proinflammatory cytokines in the lung tissue? How do they affect TLR expression?
  5. I believe the real review starts in point 3. that should be more extensive, as it is now. While points 1 and 2 should be considerably shortened and more focused. 
  6. Out of the 112 references 25 were improperly referenced. Except of the author names, date, journal etc of publication, one can only see the title and the date when the article was accessed in the search engine. It should be corrected.

Reviewer 2 Report

The presented review article is trying to summarize our current state of knowledge about the role of recently described mutual interactions between TLR and WNT signaling in immunomodulation and cancerogenesis in lung cancer. The authors discuss mainly the impacts of the TLR pathways activation by WNT ligands on the tumor microenvironment, immune responses and tumor growth. The review brings a useful piece of information, however, there are some points that should be clarified.

Specific comments:

1.       TLRs activation has different outputs in immune cells, compared to the tumour cells. In immune cells, it can activate antitumor immunity, although also immune suppression, while in tumour cells it is associated rather with direct protumorigenic effects. This should be more clarified in the text in the context of WNT signaling impacts and also possible antitumor effects should be discussed more in detail.

2.        It is not always clear in the text which data on the WNT/TLR interactions are general and which ones are related to the lung cancer. It would be useful to divide the chapter 3 into 3.1 and 3.2 like the chapters 1 and 2 are.

3.       In Fig.5 you suggest the scheme in which increased production of WNT promotes IL-6 production. However, in Oderup et al., 2013 (cited in this review) it is shown that Wnt5A, but not Wnt3A, inhibits IL-6 production. So, the effects on IL-6 should be reconsidered and clarified.

4.       In your schemes, IL-6 plays a central role as a proinflammatory cytokine. You should also discuss its crucial role in tumorigenesis and immune suppression.

Round 2

Reviewer 1 Report

The manuscript has improved. It is recommended that a native English speaker reads it once more. 

Reviewer 2 Report

The manuscript has been substantially completed and improved. There are some typos appearing in the new parts (e.g. line 164 “protumral by promoting..”, line 170 “TLR5 has been report…”), so it should be double-checked before publishing.